



# Integrated management of a Swiss cropland is not sufficient to preserve its soil carbon pool in the long-term

Carmen Emmel[1], Annina Winkler[1], Lukas Hörtnagl[1], Andrew Revill[1,2], Christof Ammann[3],
Petra D'Odorico[1,4], Nina Buchmann[1], and Werner Eugster[1]

[1]ETH Zurich, Department of Environmental Systems Science, Institute of Agricultural Sciences, 8092 Zurich, Switzerland
[2]School of GeoSciences, University of Edinburgh, Edinburgh, United Kingdom
[3]Agroscope, Federal Research Station, Climate and Air Pollution, 8046 Zurich, Switzerland
[4]Department of Biology, University of Toronto at Mississauga, Mississauga, ON, L5L 1C6, Canada

**Correspondence:** Carmen Emmel (carmen.emmel@usys.ethz.ch)

**Abstract.** Croplands are involved in the exchange of carbon dioxide ($CO_2$) between the atmosphere and the biosphere. Furthermore, soil carbon (C) stocks play an important role in soil fertility. It is, thus, of great interest to know whether croplands act as a net source or sink of atmospheric $CO_2$, and if soil C stocks are preserved over long timescales. The FLUXNET site CH-Oe2 in Oensingen, Switzerland has been operational since the end of 2003. This cropland is managed under the Swiss framework of

the Proof of Ecological Performance (PEP, a variant of integrated management) with a crop rotation centred on winter wheat, which also includes winter barley, winter rapeseed, peas, potato and intermediate cover crops. In addition to eddy covariance measurements, meteorological and soil measurements were available along with information on C imports and exports from organic fertilisation, sowing and harvesting. This study investigates cropland C budgets over 13 years and assesses whether the PEP regulations resulted in a balanced C budget. Strongest $CO_2$ uptake was observed during cereal seasons. C export through

harvest, however, offset the strong uptake of the cereal crops. The largest net $CO_2$ emissions to the atmosphere were observed during pea and cover crop seasons. Net biome production, representing the overall C budget, typically ranged between close to C neutral to C losses of up to 407 g C m$^{-2}$ per season, with peas being the largest source. Overall, the field lost 1674 g C m$^{-2}$ over thirteen years (129 g C m$^{-2}$ yr$^{-1}$), which was confirmed by soil C stock measurements at the beginning and the end of the study period. Although managing the field under the regulations of PEP did not result in an overall C sink, model simulations

showed that the use of cover crops reduced the C losses compared to leaving the field bare. The use of solid manure improved the C budget by importing substantial amounts of C into the soil while liquid manure had only a small effect. We thus conclude that additional efforts are needed to bring Swiss management practices closer to the goal of preserving soil C in the long-term.





## 1    Introduction

The net carbon (C) exchange of agricultural fields, which are typically highly managed, is of interest in the context of global warming and rising atmospheric carbon dioxide ($CO_2$) concentrations (Ciais et al., 2013). Through photosynthesis, $CO_2$ is removed from the atmosphere, whilst respiration of soils and plants releases $CO_2$ to the atmosphere. An ecosystem can be a

net $CO_2$ source or sink from an atmospheric point of view, depending on whether photosynthesis or respiration dominates. This exchange of $CO_2$ between an ecosystem and the atmosphere is typically measured with the eddy covariance technique (Baldocchi, 2003; Eugster and Merbold, 2015) as net ecosystem exchange (NEE).

Soil C concentrations have an important influence on soil fertility by improving the soil water holding capacity, nutrient storage, aggregation and sorption of organic or inorganic pollutants (Smith et al., 2015). Because agricultural land makes up

approximately 37 % of the world's land surface (The World Bank, 2017) and holds substantial amounts of C, soil management can be a powerful means of mitigating C losses of croplands (Lal et al., 2011). Therefore, it is of great interest to determine whether agricultural ecosystems are a C source over longer timescales and how this influences the C stocks in the soil.

To understand whether an ecosystem is losing C, all exports (e.g., harvests) and all imports (e.g., organic fertilisers or seeds) of C need to be considered in order to calculate the net biome production (NBP). There have been a number of studies

investigating NEE and the C budget of different ecosystems, however, most of them focused on forests (e.g., Turner et al., 1995; Etzold et al., 2010; Adachi et al., 2011; Zielis et al., 2014) and grassland ecosystems (e.g., Allard et al., 2007; Ammann et al., 2007; Gilmanov et al., 2007; Soussana et al., 2007; Li et al., 2008). Long-term cropland flux stations are relatively few resulting in a much lower number of cropland studies. In contrast to forested ecosystems, croplands are often considered overall C sources (Ceschia et al., 2010). Schulze et al. (2009) for example reported a significant source of 33 Tg C yr$^{-1}$ for Continental

European croplands. This may lead to a strong decrease in soil C because large amounts of photosynthetically-fixed $CO_2$ are removed from the field during harvest and only a relatively small amount of biomass, in form of residues and litter, is returned to the soil (Janzen, 2006). The management type and intensity of agricultural ecosystems strongly influences the net C budget (Ceschia et al., 2010; Eugster et al., 2010). Some studies have found that croplands growing specific crops (e.g., maize) and/or under specific management practices (e.g., no tillage or reduced tillage) were net C sinks or C neutral (e.g., Hollinger et al.,

2005; Nishimura et al., 2008; Robertson et al., 2000).

Most of the existing cropland studies looked either at short periods of measurements (single years to only one crop rotation) from single field sites (e.g., Anthoni et al., 2004; Moureaux et al., 2006, 2008; Aubinet et al., 2009; Béziat et al., 2009; Schmidt et al., 2012; Chi et al., 2016), combined measurements from different field sites (Janssens et al., 2003; Ceschia et al., 2010; Eugster et al., 2010; Kutsch et al., 2010; Gilmanov et al., 2013; Joo et al., 2016; Jensen et al., 2017) or were based on model

simulations (e.g., Parazoo et al., 2014; Vuichard et al., 2016). Prescher et al. (2010) pointed out the need of long periods for investigating management influences on NBP. Furthermore, only with long-term measurements a direct comparison with soil C stocks can be made, because stocks change only slowly and are typically only measured at decadal intervals. There have been only three studies analyzing the C budget of croplands in detail at a single site over a longer timescale: Suyker and Verma





(2012) and Dold et al. (2017) studied maize-soybean rotations in the United States over eight and nine years, respectively, and Buysse et al. (2017) studied a four-year crop rotation field in Belgium over twelve years.

At the Swiss FluxNet cropland site CH-Oe2 in Oensingen, Switzerland, long-term eddy covariance and meteorological measurements have been conducted since 2003. This is the only long-term Swiss FluxNet cropland site. The field is managed

under the Swiss integrated management framework of the Proof of Ecological Performance (PEP) (Swiss Federal Council, 2017). The term "integrated management" is here defined as a more sustainable management approach when compared to conventional agricultural practices and does not only focus on economical benefits but also takes ecological aspects into account. These agricultural regulations were introduced in Switzerland in the late 1980s. The PEP regulations include amongst other requirements, the fulfilment of neutral nitrogen (N) and phosphorus budgets, the implementation of a crop rotation, an

appropriate soil protection (e.g., by planting cover crops in the autumn, to avoid bare fields during winter), and the reduction and more appropriate use of fertilisers and pesticides.

Given that there is little known about the detailed long-term C budgets of crop fields, especially in Switzerland, and to understand whether implementing PEP has also led to a balanced C budget, the objectives of this study were to (1) analyse NBP of the crop field over thirteen years, (2) determine the impact of the different crop types on NBP, and (3) assess the

differences in C loss by planting a cover crop compared to a bare field.

## 2   Material and methods

### 2.1   Measurement site

The CH-Oe2 field site is located in Oensingen, in the canton of Solothurn, Switzerland ($47°17'11.1''$N, $7°44'01.5''$E, 452 m a.s.l.). The crop field has an extend of 1.55 ha with a fluvisol with 42 % clay, 33 % silt and 25 % sand (Alaoui and Goetz,

2008). The average air temperature ($T_A$) at the site is 9.8 °C, and the average annual precipitation sum ($Prec$) is 1155 mm (Fig. 1, period 2004 to 2016; the diagram was produced in $R$ with the diagwl function of the climatol package). The field has been managed under the regulations of PEP since the late 1990s, featuring a three-year crop rotation (Table 1). The main crop has been winter wheat, which is usually planted every third year followed by winter barley. The third crop in the rotation was either potato, winter rapeseed or peas. Only between autumn 2006 and autumn of 2010, wheat was planted every second

year. Before summer crops (potato or peas) were sown, a mixture of summer oat, *Phacelia*, and Alexandrine clover (2005) or *Phacelia* only was planted (2009 and 2015). After every rapeseed harvest, a voluntary regrowth of the rapeseed was allowed and the newly grown rapeseed plants were then mulched and incorporated into the soil later in the autumn before wheat was sown. Before the management under PEP started in the late 1990s, the field had an eight-year arable-ley rotation, including three years of perennial grass-clover mixture.

Management information including dates and type of tillage, sowing dates and seed weights, fertilisation dates and amounts, dates of pesticide applications as well as harvest dates and yield (grain and straw) was regularly provided by the farmer (Table 1). Management timing and field conditions were confirmed with webcam images of the field (since 20 May 2005 taken at 10:30, 12:30 and 14:30 CET (UTC + 1 hour) and since 01 March 2015 at 9:30, 12:30 and 14:30 CET; until 18 December





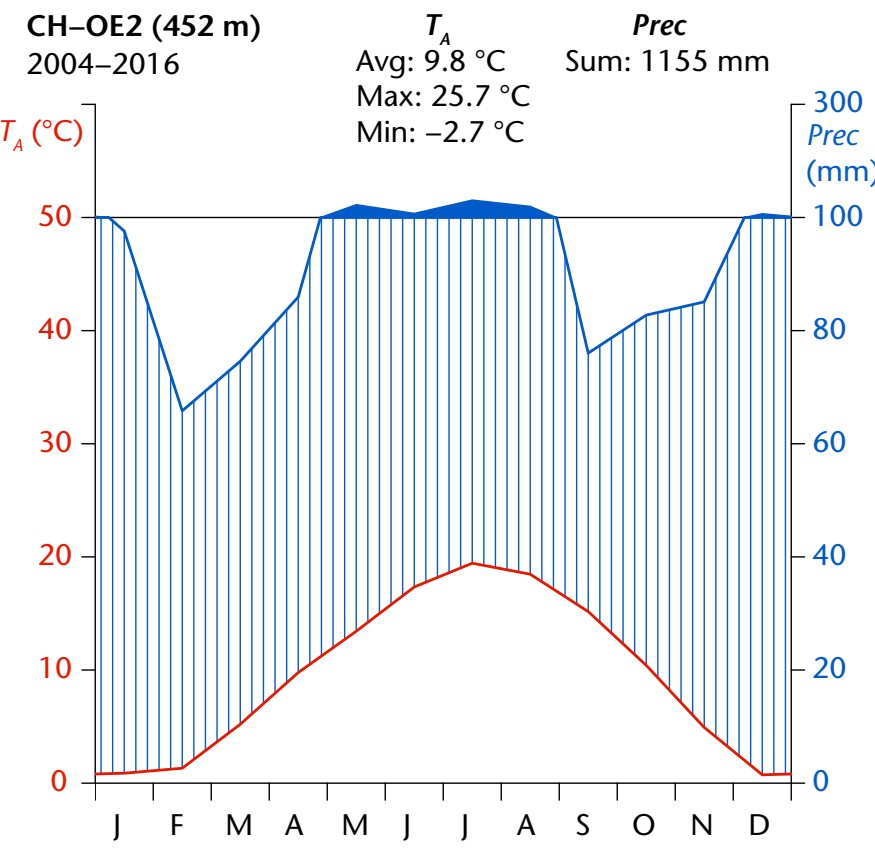

**Figure 1.** Climate diagram after Walter and Lieth (1960) for the time period of 2004 to 2016. The monthly average air temperature ($T_A$) and the monthly total precipitation ($Prec$) are shown in red and blue, respectively. Average (Avg), average minimum (Min) and average maximum (Max) annual $T_A$ and average annual total (Sum) $Prec$ are listed at the top of the figure. Note that the scale of the right axis changes above 100 mm.





2013, DCS-900, D-Link, Taipeh, Taiwan and afterwards NetCam SC 5MP, Stardot, Buena Park, California, USA). In the case of wheat, barley and rapeseed, the moisture content of the harvested grains was reported by the farmer. Cover crops were not harvested and, thus, ploughed into the soil. No harvest was conducted for the potatoes in 2006; due to a hail storm on 05 July 2006 the potatoes were of very poor quality, therefore left in the ground and later ploughed under. Between 2004 and 2016,

solid manure was applied on three occasions (always at the end of the cover crop seasons), whereas liquid manure was applied on five occasions (at the end of wheat, barley and rapeseed seasons). A crop season is here defined as the period between sowing of a crop and sowing of the following crop. Mineral fertilisers were applied during all crop seasons, except for cover crops and the 2016 pea season. Herbicides were applied during all crop seasons, except for cover crops and the potato season. Fungicides were only used in spring 2004 (wheat), 2005, 2012 and 2015 (all barley), and insecticides were applied during

the rapeseed season in 2007/2008 as well as during the 2010 pea season. Grubbing (shallow secondary tillage) was conducted almost every year, ploughing approximately every third year (typical depth of 30 cm), and harrowing was conducted every year since 2010. In 2005, the cover crop was mulched on 09 November. In 2010 and 2016, the cover crop was incorporated into the ground shortly before the next crop was sown without any preceding mulching.

## 2.2   Turbulent fluxes

Since the end of December 2003, eddy covariance (EC) measurements have been made at the site. The eddy covariance measurements consist of three-dimensional wind speed and air temperature measurements with an ultrasonic anemometer (R3-50, Gill Instruments Ltd., Lymington, Hampshire, UK) as well as $CO_2$ and water vapour measurements with an open-path infrared gas analyser (LI-7500, LI-COR, Lincoln, NB, USA) and were recorded at 20 Hz.

The eddy covariance data were processed and quality controlled with the software EddyPro (Version 6.2.0, LI-COR).

Thereby 30-min averaged fluxes were calculated and the following corrections and filters were applied: high-frequency despiking and a drop out test (on the raw data) following Vickers and Mahrt (1997), angle-of-attack correction (Nakai et al., 2006), double rotation (Wilczak et al., 2001), lag time compensation via covariance maximisation using a default lag time if a maximum was not attained within a plausible window, density fluctuation correction (Webb et al., 1980), high-pass filter (Horst, 1997), low-pass filter (Moncrieff et al., 2004) as well as a steady state test and test for well developed turbulence con-

ditions (on the processed fluxes). Fluxes were rejected from further analyses when they were outside a physically plausible range ($\pm 50$ $\mu$mol m$^{-2}$ s$^{-1}$). From November 2015 to May 2016, an angle-of-attack filter was also applied, which discarded half-hourly fluxes if the angle of attack was outside the range of -10 to 30° for more than 10% of the half hours. This additional quality criterion was applied to filter out time periods of an occasional malfunctioning of an anemometer transducer. During times of repair of the R3-50 ultrasonic anemometer, the ultrasonic anemometer was replaced by a model HR-100 ultrasonic

anemometer (Gill).

NEE was calculated by adding the half-hourly $CO_2$ flux and $CO_2$ storage and subsequently despiked by iteratively removing outliers outside the valid range defined as the mean $\pm$ three times its standard deviation (Rogiers et al., 2004) based on a 30-day moving window. NEE was then gap filled and partitioned into gross primary production (GPP) and ecosystem respiration ($R_{eco}$) based on Reichstein et al. (2005) using the $R$ software REddyProc by the MPI Jena (Version 1.0.0., Reichstein et al.,



**Table 1.** Management information for all 16 crop seasons (defined as sowing of the current crop to sowing of the following crop) between 2003 and 2016 with crop type, sowing and harvest dates and yield (G = grain, P = peas, S = straw). Moisture content of the harvested biomass (MC) is given in brackets. If manure was applied during the crop season, date, manure type and amount are given as well.

| Crop | Sowing | Harvest | | Manure application | |
|---|---|---|---|---|---|
| | | Date | Yield, kg ha$^{-1}$ (MC, %) | Date | Type (amount) |
| Wheat | 16 Oct 2003 | 04 Aug 2004 | G: 7980 (13.7), S: 4030 (11.1) | – | – |
| Barley | 29 Sep 2004 | 14 Jul 2005 | G: 6940 (12.3), S: 1700 (11.8) | – | – |
| Cover crop | 09 Aug 2005 | not harvested | – | 24 Jan 2006 | solid (13 t) |
| Potatoe | 05 May 2006 | not harvested | – | – | – |
| Wheat | 19 Oct 2006 | 15 Jul 2007 | G: 6140 (11.8), S: 4400 (11.1) | – | – |
| Rapeseed | 28 Aug 2007 | 16 Jul 2008 | G: 3160 (5.8) | – | – |
| Wheat | 07 Oct 2008 | 21 Jul 2009 | G: 6880 (13.1), S: 3660 (11.1) | 04 Aug 2009 | liquid (33 m$^3$) |
| Cover crop | 12 Aug 2009 | not harvested | – | 06 May 2010 | solid (10 t) |
| Peas | 09 May 2010 | 19 Jul 2010 | P: 5290 (84.8) | – | – |
| Wheat | 15 Oct 2010 | 02 Aug 2011 | G: 7810 (12.8), S: 3910 (11.1) | 02 Sep 2011 | liquid (20 m$^3$) |
| Barley | 24 Sep 2011 | 09 Jul 2012 | G: 8700 (11.6), S: 2130 (11.8) | 28 Aug 2012 | liquid (30 m$^3$) |
| Rapeseed | 04 Sep 2012 | 28 Jul 2013 | G: 3920 (9.7) | 24 Sep 2013 | liquid (30 m$^3$) |
| Wheat | 19 Oct 2013 | 24 Jul 2014 | G: 7480 (16.2), S: 4400 (11.1) | 12 Sep 2014 | liquid (30 m$^3$) |
| Barley | 29 Sep 2014 | 04 Jul 2015 | G: 8110 (11.8) , S: 1580 (11.8) | – | – |
| Cover crop | 03 Aug 2015 | not harvested | – | 15 Mar 2016 | solid (20 t) |
| Peas | 09 May 2016 | 25 Jul 2016 | P: 500 (84.8) | – | – |





2017). Gap filling was done after applying an automatically determined $u_*$ filter (with a threshold ranging between 0.01 and 0.13 m s$^{-1}$; changed for each crop season). In total, NEE had to be gap filled for 46% of the half hours.

For the beginning of the first wheat season (October to December 2003), the measurement station was not established yet and therefore no flux data were available. From November 2006 until Feb 2007, no reliable NEE measurements were available

due to a sonic anemometer malfunctioning. Therefore, NEE was estimated for these two time periods in 2003 and 2006/2007 by averaging gap-filled NEE of the corresponding days of the wheat seasons in 2008, 2010 and 2013 (on a daily basis).

### 2.2.1 Yield, seed and manure

Moisture contents of straw and seeds were determined in the lab by weighing a subsample with a high precision scale before and after drying in the oven at 55 °C. Elemental C concentrations of dried and ground yield as well as seed samples were measured

with a Flash EA 1112 Series elemental analyser (Thermo Italy, Rhodano, Italy) coupled to a Finnigan MAT DeltaplusXP isotope ratio mass spectrometer (Finnigan MAT, Bremen, Germany) according to Brooks et al. (2003) and Werner et al. (1999), with a sample, blank and laboratory standard positioning (Identical-Treatment principle) following Werner and Brand (2001). The performance was tested with laboratory standards. The C concentrations and moisture contents of manure were measured in 2006 (solid) and 2017 (liquid) at the laboratory LBU (Thun, Switzerland) and in 2009 (liquid) by Agroscope (Zurich,

Switzerland). The measurements in 2006 were used for all other solid manure applications (2006, 2010 and 2015) as well. In the case of liquid manure, an average of all available liquid manure measurements of CH-Oe2 and the neighbouring site CH-Oe1 (same farm, 2002-2011; Ammann et al., 2009) were averaged when the manure was not analysed during a given year. In cases when the moisture content or C concentration of the harvested biomass were not measured, the value was substituted by the average of all other available seasons of the same crop. In the case of peas, a sample from a neighbouring field in 2017

was used to determine the moisture content of the peas at harvest. To determine the C export and import (g C m$^{-2}$) through harvest, fertilisation and sowing, first the dry weight of the yields, fertilisers and seeds was calculated and then multiplied by the corresponding C concentration.

### 2.2.2 Soil carbon and nitrogen

Soil C and N concentrations were measured in 2004 and 2017. On 13 October 2004, soil samples were taken to a depth of

12 cm at 36 locations in the field. Each sample was divided into two parts (0-6 cm, 6-12 cm depth), from which the C and N concentrations were determined with an elemental analyser (LECO CHN-1000, LECO Corp., St. Joseph, MI, U.S.A.) after sieving (1 mm mesh), drying and grinding the soil. Additionally, bulk density of the soil was determined at the same 36 locations on the field.

In 2017, soil samples for C and N measurements were taken on five days (23 February, 23 March, 05 April, 04 May and 31

May), of which two days were before and three were after the application of liquid manure (31 March 2017). At 12 locations, the samples were taken to a depth of 30 cm, and at four locations to a depth of 70 cm. These samples were divided into subsamples of $0-15$ cm, $15-30$ cm, $30-50$ cm and $50-70$ cm depth on the first 4 sample days, and $0-2$ cm, $2-5$ cm, $5-10$ cm, $10-15$ cm, $15-30$ cm, $30-50$ cm and $50-70$ cm on the last sample day. All samples were processed the same way as in



2004. Concentrations of C and N were determined with the same set up as for yield C concentrations. In 2017, bulk density of the soil was determined at four locations for $5.5-9.5$ cm depth and $20.5-24.5$ cm depth and at one location for $0.5-4.5$ cm, $5.5-9.5$ cm, $10.5-14.5$ cm, $20.5-24.5$ cm, $38.0-42.0$ cm and $58.0-62.0$ cm depth. Averages of C and N concentration and bulk density for each depth layer and year were calculated and soil C and N densities ($\rho_C$ and $\rho_N$, respectively) were

then determined by multiplying the average C and N concentration of a depth layer by the corresponding average bulk density. For stock calculations, the $\rho_C$ or $\rho_N$ of each depth layer was multiplied by the layer thickness and then all depth layers were summed.

The uncertainty of the LECO CHN 1000 analyser was determined from repeated measurements of two standards and one blank (standard deviations). At concentrations in the range of soil samples the accuracy of the C and N contents is $\pm 1.7$ % and

$\pm 3.9$ %, respectively. The uncertainty of the C and N contents measured with the elemental analyser in 2017 were $\pm 1.5$ % and $\pm 1.7$ % of the C and N contents, respectively, determined as the average from 7 batches

### 2.2.3 Ancillary meteorological and soil measurements

Further ancillary meteorological and soil measurements have been made at the site since the end of 2003. The set up consists of an air temperature and relative humidity sensor (CS215, Campbell Scienctific Ltd., Logan UT, USA; 2 m height), a cup

anemometer (A100R, Vector Instruments, Denbighshire, UK; 2 m height) and a wind vane (W100P, Vector Instruments; 2 m height), a four-component net radiometer (CNR1, Kipp & Zonen, Delft, The Netherlands; until November 2014 at 1 m height, afterwards at 2 m height), a sunshine sensor measuring diffuse and total photosynthetically active radiation (until June 2014 BF3, afterwards BF5, Delta T, Cambridge, UK; until November 2014 at 1 m height, afterwards at 2 m height), four heat flux plates (HFP01, Hukseflux B.V., Delft, The Netherlands; 0.03 m depth) with corresponding soil temperature probes (model

107, Campbell Scientific; 0.015 m depth), a soil moisture probe profile (ECH2O, Decagon Devices Inc., Pullmann, WA, USA; 0.05, 0.15, 0.30, 0.50 m depths), a soil temperature profile (Th3-s, UMS GmbH, Munich, Germany; 0.05, 0.10, 0.20, 0.30, 0.50 and 1.00 m depths) and a heated rain gauge (until July 2014 model 10116, Toss GmbH, Potsdam, Germany, afterwards model 15188, Lambrecht GmbH, Göttingen, Germany; 1 m height). These measurements were conducted at a frequency of 1 Hz and 30-min averaged until October 2012. Afterwards 1-min averages were recorded. These data, aggregated to 30-minute

resolution, were used to conduct the flux data gap filling and partitioning, to run the SPA-Crop model (Section 2.2.5).

### 2.2.4 Estimation of net biome production

NBP was used to determine the C budget of the field between 2003 and 2016. Knowing the C exchange through turbulent $CO_2$ fluxes (NEE), C export by harvest ($E_{harvest}$) and C imports by organic fertiliser ($I_{fertiliser}$) and sowing ($I_{sowing}$), NBP can be calculated as:

$$\text{NBP} = \text{NEE} + E_{harvest} + I_{fertiliser} + I_{sowing} \tag{1}$$

We use the same sign convention as Buysse et al. (2017): when the field is a C source, NBP is positive, while it is negative if it is a C sink. For the contributing terms, C imports into the ecosystem are negative and exports positive. This C budget only



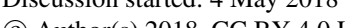


includes $CO_2$ fluxes and neglects methane fluxes, which tend to be very small for crop fields other than rice paddies (e.g., Nishimura et al., 2008), additional volatile organic compounds (VOC) and dissolved organic carbon losses (DOC). Cumulative NBP ($NBP_{cum}$) is then defined as:

$$NBP_{cum} = \int_{t_0}^{t} NEE + \int_{t_0}^{t} E_{harvest} + \int_{t_0}^{t} I_{fertiliser} + \int_{t_0}^{t} I_{sowing}, \tag{2}$$

5    , where $t_0$ and $t$ are the starting and end dates of the period of interest, respectively. The first term of this equation is the cumulative NEE ($NEE_{cum}$).

### 2.2.5   Modelled net ecosystem exchange

In order to quantify the impact of the cover crop on the C budget, the Soil-Plant-Atmosphere Crop (SPA-Crop, Sus et al., 2010) was used to simulate NEE under the same meteorological conditions but without the cover crop (i.e., bare soil). The model

10   simulates cropland ecosystem photosynthesis and water-balance at point-scales over fine temporal (half-hourly) and vertical scales (ten canopy and twenty soil layers). The SPA-Crop simulation of heterotrophic respiration, modelled independently of crop type, includes decomposing surface litter and soil organic C (SOC) pools. The simulations were applied for the three available cover crop periods by running the model for the entire previous year (not shown) until the end of the cover crop season. The results were then compared to the corresponding eddy covariance NEE observations.



# 3 Results and discussion

## 3.1 Carbon budgets over 13 years

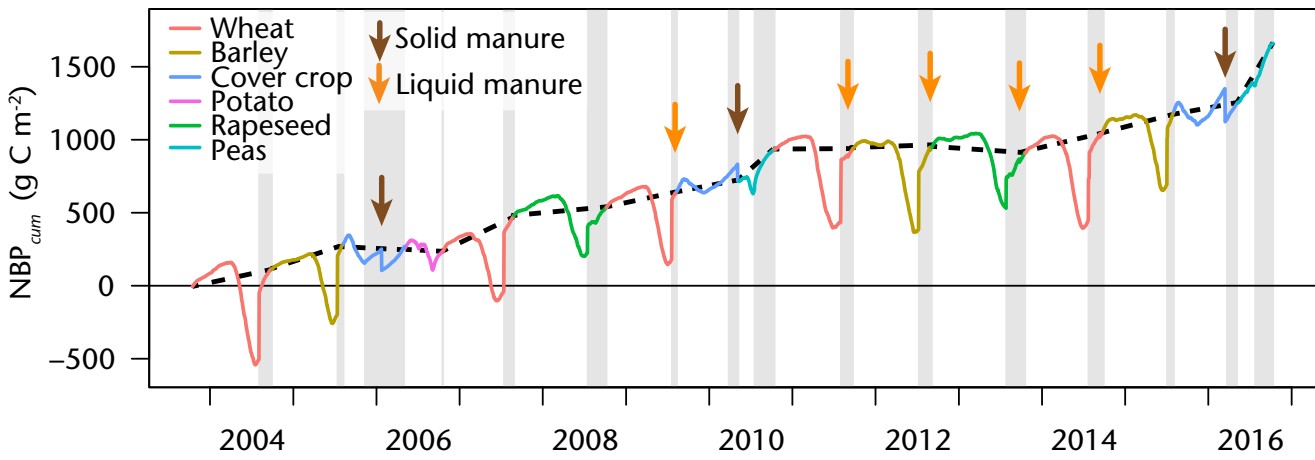

**Figure 2.** Daily cumulative net biome production ($NBP_{cum}$) between 16 Oct 2003 and 11 Oct 2016. The black dashed line connects $NBP_{cum}$ at the end of each crop season. During time periods with a grey background, the field was bare (from harvest of a crop to sowing of the next crop or from first ploughing after sowing of a crop to sowing of the next crop if the first crop was not harvested).

For all crops, the season started with a net release of $CO_2$ until the crop had emerged and became established, after which GPP began to exceed $R_{eco}$ (Fig. 2). A few weeks before harvesting, when senescence started, $R_{eco}$ exceeded GPP again, resulting

5  in a net $CO_2$ release. At the point of harvest, C was exported from the ecosystem, which can be seen in most years as a sharp increase in $NBP_{cum}$. Organic fertilisation with solid or liquid manure and sowing were C imports into the ecosystem. However, only solid manure applications were large enough C imports to be seen as a sharp decrease in $NBP_{cum}$. While the field was bare, it was almost only respiring and therefore $NBP_{cum}$ increased during these periods. The voluntary regrowth after the harvest of rapeseed (2008 and 2013) resulted in an approximately 2-month long period of uptake in the autumn of the same

10  years.

    For peas, the period of net C uptake was quite short (less than one month in contrast to three to four months for the other crops), which is due to their short growing period as they were peas for canning and were therefore harvested relatively early. The period of net C uptake is barely visible during the pea season in 2016 because the field was flooded due to extensive rain. Cover crops were only growing in the autumn resulting in a relatively weak $CO_2$ uptake followed by a relatively long period

15  of net $CO_2$ loss.

    $NBP_{cum}$ at the end of each season mostly increased over time. Only during the potato season in 2006 without harvest due to the hail damage and during the crop rotation cycle (wheat, barley and rapeseed) between 2010 and 2013, $NBP_{cum}$ stayed almost constant. $NBP_{cum}$ of the first crop rotation cycle (wheat, barley, cover crop, potato; $2003 - 2006$) was 236 g C m$^{-2}$.



Between 2006 and 2009, wheat was repeated every second year. During the first two-year period (wheat, rapeseed), the field was a net source of 302 g C $m^{-2}$ and during the second two-year period (wheat, cover crop, peas) a net source of 396 g C $m^{-2}$. During the next full crop rotation cycle (wheat, barley, rapeseed; $2010 - 2013$), the field was close to C neutral (NBP = $-22$ g C $m^{-2}$), while it was a net source of 748 g C $m^{-2}$ during the last crop rotation (wheat, barley, cover crop, peas; $2014 - 2016$).

The cumulative net biome production ($NBP_{cum}$) for the 16 crop seasons between autumn 2003 and autumn 2016 shows that there was a net C loss of 1674 g C $m^{-2}$ over the thirteen years of study (Table A2). The field lost on average $129 \pm 50$ g C $m^{-2}$ of C per year (unless stated otherwise, we report mean $\pm$ standard error except for soil C and N values, where mean $\pm$ standard deviation is given).

Soil C densities in the top 12 cm of the field were $0.0356 \pm 0.0042$ g $cm^{-3}$ (mean $\pm$ standard deviation) in 2004 and
decreased on average by 18.5 % (P value < 0.001) to $0.0290 \pm 0.0045$ g $cm^{-3}$ until spring 2017 (average over the top 15 cm and over all measurement days in 2017). The bulk density of the same layer increased from $1.16 \pm 0.08$ g $cm^{-3}$ in 2004 to $1.21 \pm 0.14$ g $cm^{-3}$ in 2017. The C soil stock decreased on average by 790 g C $m^{-2}$ in the top 12 cm from $4270 \pm 506$ g C $m^{-2}$ to $3480 \pm 540$ g C $m^{-2}$. At the same time, N stock changes were not significant over the thirteen years ($0.00311 \pm 0.00043$ in 2004, $0.00320 \pm 0.00051$ g N $cm^{-3}$ in 2017, P value = 0.5217). There were no measurements from deeper soil layers available
for 2004. However, measurements in 2017 show that C densities did not vary significantly in the top 30 cm (Fig. 3). Also ploughing was done in most years to a depth of 30 cm. If we therefore assume that C stocks changed equally over a depth of 30 cm between 2004 and 2017, the soil C stock decreased in the top 30 cm layer on average by 1980 g C $m^{-2}$. This corresponds to an annual average loss of 152 g C $m^{-2}$.

The application of slurry caused such a small C input that it was not only invisible in $NBP_{cum}$ (Fig. 2) but was also not
detectable in the soil. Soil C density ($\rho_C$) measurements before and after the application of the slurry in 2017 did not reveal any significant changes (Fig. B1). The slurry added only 25.4 g C $m^{-2}$ and 4.7 g N $m^{-2}$ to the soil. When comparing these numbers to the C and N stock of the top 30 cm of the soil, it can be seen that the C and N input is negligible.



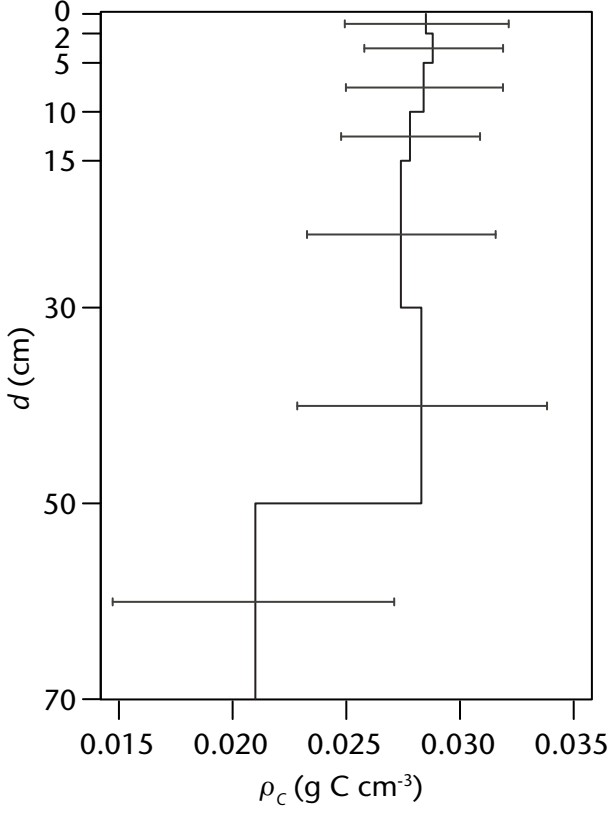

**Figure 3.** Average soil carbon density ($\rho_c$) for seven different depth layers between the surface and a depth of $d = 70$ cm (31 May 2017). Error bars show standard deviations.

The field site was clearly a C source, which was also confirmed by the changes in soil C stocks measured at the beginning and at the end of the measurement period. Depending on the measurement method, the field lost 15.7 % (based on $\text{NBP}_{cum}$ and soil C stock of top 30 cm in 2004) to 18.5 % (based on soil C stocks in 2004 and 2017) of C over the thirteen years. The differences between the C budget determined by calculating NBP and by measuring C stocks in the soil were remarkably small given that these results are based on two completely independent measurements.

Ceschia et al. (2010) studied the C exchange ($138 \pm 239$ g C m$^{-2}$) and the changes in soil C stocks of the top 30 cm ($2.4 \pm 4.7$ %) of European croplands (averaging over 17 croplands and 41 site years $\pm$ standard deviation, between 1 and 5 consecutive years per site). In contrast to our results, their findings were not significantly different from a C neutral budget. However, our results were within the range found by Ceschia et al. (2010). There are a number of other studies on European crop fields with crop rotations that found similar or slightly higher annual losses to what we found in this study (e.g., Prescher et al., 2010; Buysse et al., 2017, no cover crops included in these studies). A modelling approach based on soil stock measurements for European croplands also resulted in comparable average annual C losses of approximately $90 \pm 50$ g C m$^{-2}$ (Janssens



et al., 2003). The management under the regulations of PEP did not result in a neutral C budget or C sink and also not in a significantly smaller average annual loss compared to other European croplands. However, soil N stock measurements showed that the neutral N budget as required by PEP was approximately reached.

Several studies have looked at the uncertainties of the contributing terms of NBP. Aubinet et al. (2009) reported an NBP uncertainty of 140 g C m$^{-2}$ over a full crop rotation of four years (at NBP = 220 g C m$^{-2}$). Buysse et al. (2017) listed in detail the uncertainties involved in the different NBP terms, which would add up to an uncertainty of 220 g C m$^{-2}$ over the 12 years of their study (at NBP = 990 g C m$^{-2}$). They determined uncertainties due to C export with harvest and import with organic fertilisers to be the main contributors to the overall NBP uncertainty, which was also found by Béziat et al. (2009). Uncertainties in our study can be assumed similar.

## 3.2 Crop specific budgets

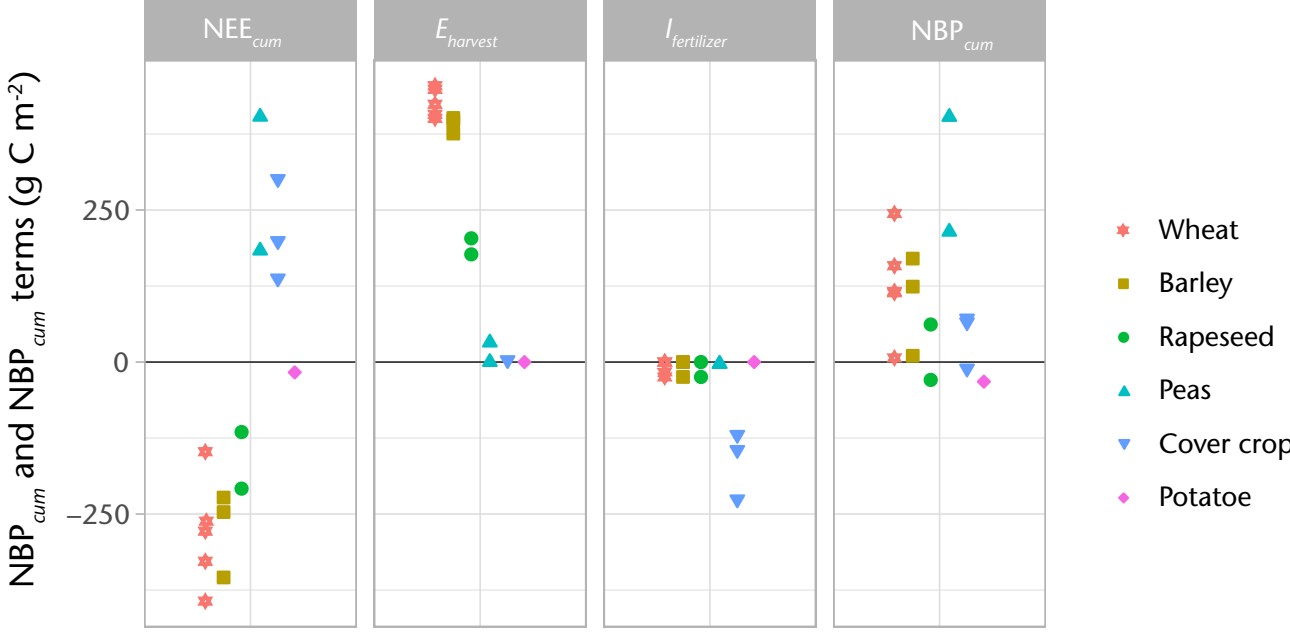

**Figure 4.** Crop season specific cumulative net biome productions (NBP$_{cum}$) and their main contributing terms: cumulative net ecosystem exchange (NEE$_{cum}$), C export by harvest ($E_{harvest}$) and C import by fertiliser ($I_{fertiliser}$). Each symbol stands for one crop season. Not shown is the C import by sowing ($I_{sowing}$), which is negligibly small except for potatoes. It is however included in the calculation of NBP$_{cum}$.

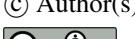

**Table 2.** Average and standard error of cumulative net ecosystem exchange ($NEE_{cum}$), C export through harvest ($E_{harvest}$) and cumulative net biome production ($NBP_{cum}$) in g C m$^{-2}$ season$^{-2}$ of the five crop types with more than one season. The number ($n$) of seasons for each crop type is given in brackets.

|  | $NEE_{cum}$ | $E_{harvest}$ | $NBP_{cum}$ |
|---|---|---|---|
| Wheat ($n = 5$) | $-284 \pm 50$ | $427 \pm 12$ | $130 \pm 49$ |
| Barley ($n = 3$) | $-279 \pm 41$ | $391 \pm 8$ | $98 \pm 49$ |
| Rapeseed ($n = 2$) | $-165 \pm 47$ | $191 \pm 14$ | $13 \pm 46$ |
| Cover ($n = 3$) | $205 \pm 47$ | $0$ | $38 \pm 28$ |
| Peas ($n = 2$) | $296 \pm 112$ | $19 \pm 16$ | $311 \pm 96$ |

Wheat and barley showed the largest net C uptake from the atmosphere over the crop season and had also the largest C export through harvest (Fig. 4 and Table 2). They were followed by rapeseed, for which also less C was exported with the harvest. Peas and cover crops assimilated less C from the atmosphere than the ecosystem released at the same time. In the case of peas very little was exported from the field during harvest. For all crops, $I_{sowing}$ was very small to negligible ($< 15$ g C m$^{-2}$, Table

A1). Also the application of slurry resulted in rather small imports of 16 to 25 g C m$^{-2}$, whereas solid manure imported 123 to 229 g C m$^{-2}$ (Table 1 and A1).

Taking into account $NEE_{cum}$, $E_{harvest}$, $I_{fertiliser}$ and $I_{sowing}$, $NBP_{cum}$ of most crop seasons was positive. Pea seasons showed a substantially larger overall C loss than the other crops. Most other crop seasons ranged between close to zero and 160 g C m$^{-2}$. The potatoes not being harvested and not receiving a fertiliser application resulted in the one available season in an

almost neutral C budget. For one season (2007/08) rapeseed was a weak C source and a very weak C sink in the other season (2012/13). Cover crops were only on the field from the late autumn until the early spring and had thus less light available and grew usually under colder conditions than the other crops. Their relatively large C loss to the atmosphere was strongly compensated by the application of solid manure. Solid manure was always applied at the end of the cover crop seasons. This was done to compensate the expected C losses during the following pea season, which are often referred to by farmers as

consumers of soil organic C. Therefore, it could be argued that the application of solid manure should be attributed to the following pea season instead of the cover crop season. With the crop season defined as the time range between first ploughing after the harvest of the previous crop to the first ploughing after the harvest of the current crop, all crops (except potatoes and one barley season) would be in a more similar range (peas: 124 g C m$^{-2}$ in 2010 and 181 g C m$^{-2}$ in 2016; Fig C1). The attribution of the manure application to the pea season is also discussed in Gilmanov et al. (2014). The reduction of the net C

loss during the pea season due to the solid manure application shows that the application of solid manure before the growth of peas is useful to compensate the loss of C during these seasons although it can only partly offset the C losses.

Our results for winter wheat and winter barley are comparable to what was found in Europe for these crop types (averaged over several sites, seasonal $NEE_{cum} = -304 \pm 49$ and $-303 \pm 92$ g C m$^{-2}$, $E_{harvest} = 513 \pm 44$ and $378 \pm 71$ g C m$^{-2}$, NBP $= 191 \pm 58$ and $101 \pm 104$ g C m$^{-2}$, $n = 12$ and 3, respectively; Ceschia et al., 2010). There are very few studies looking at





rapeseed or peas. For winter rapeseed (in Germany) and peas (in France), Ceschia et al. (2010) reported values of NEE = $-306$ and 278 g C m$^{-2}$, $E_{harvest}$ = 560 and 98 g C m$^{-2}$ and NBP = $-2$ and 375 g C m$^{-2}$, respectively, including only one season per crop type. In our study rapeseed assimilated less C in both seasons and also less C was exported with the harvest, however, NBP was again comparable. For peas, NEE and NBP was comparable but less was exported with the harvest at our field. This

could be related to the fact that the peas cultivated at CH-Oe2 were peas for canning, which are harvested when they are still relatively small. We are not aware of a study having investigated the C budget of potatoes that does not use data from our own site. The results of our potato season should not be considered representative for regular potato seasons due to the hail damage, which had major impacts on the management, the growth of the plants and resulted in no harvest. In our study, applying solid manure to the cropland was found to import substantial amounts of C to the ecosystem while the import through liquid manure

was very small. For a variety of European croplands, Ceschia et al. (2010) found that organic fertilisation tended to lower the C budget even though respiratory losses can slightly increase (less than 10 %) in the first month after the application of solid manure (Eugster et al., 2010).

### 3.3 The effect of cover crops

During the cover crop seasons there was always a net C loss, i.e. cover crop NBP was positive in all seasons. This loss,

however, could have been larger not having a crop on the field at all. Having a crop on the field, allows C uptake through photosynthesis, however, also autotrophic respiration (by the plants) and heterotrophic respiration (by providing more soil C matter to decompose) will be enhanced. Depending on whether photosynthesis or respiration is enhanced more, a cover crop is beneficial in the context of the C budget or not. In order to asses the benefit of having a cover crop, the CO$_2$ exchange of the field without crop (i.e. bare field) was modelled with the SPA-Crop model. All other terms of NBP were kept constant

since the cover crop was not harvested. SPA-Crop captures the CO$_2$ exchange from harvest of the previous crop until start of the cover crop growth quite well (Fig. 5). In all three seasons the field with cover crop is overall a smaller net C source than the bare field, even though the NEE$_{cum}$ difference covers a large range of 11 to 163 g C m$^{-2}$. The cover crop seems to be clearly beneficial (GPP increases larger than R$_{eco}$ increases) to reduce C losses during fallow periods. Furthermore, substantial amounts of C are introduced into the soil by incorporating the biomass at the end of the season when the field is prepared for

the next crop. Ceschia et al. (2010) report that also the voluntary regrowth of seeds and weeds after the harvesting of winter wheat at Avignon in the season 2005/2006 reduced the C losses. This result on cover crops shows that the regulations of PEP requiring a cover crop during fallow periods improved the C budget of the field.

### 3.4 Solid manure can at least partly compensate the C losses

The more frequent use of solid manure could compensate at least partly the C losses of the crop field. Assuming the same

average C loss rate for the future but without any organic fertiliser application (also no slurry), the average annual loss would be 174 g C m$^{-2}$. However, Switzerlands nationally determined contribution (NDC) to the reduction in greenhouse gas emissions assumes zero emissions from non-forest lands like croplands (NDC, 2017). The average C concentrations in solid manure at CH-Oe2 was 440 g kg$^{-1}$ dry mass (Table D1). Based on these numbers an annual manure application of approximately 15.8





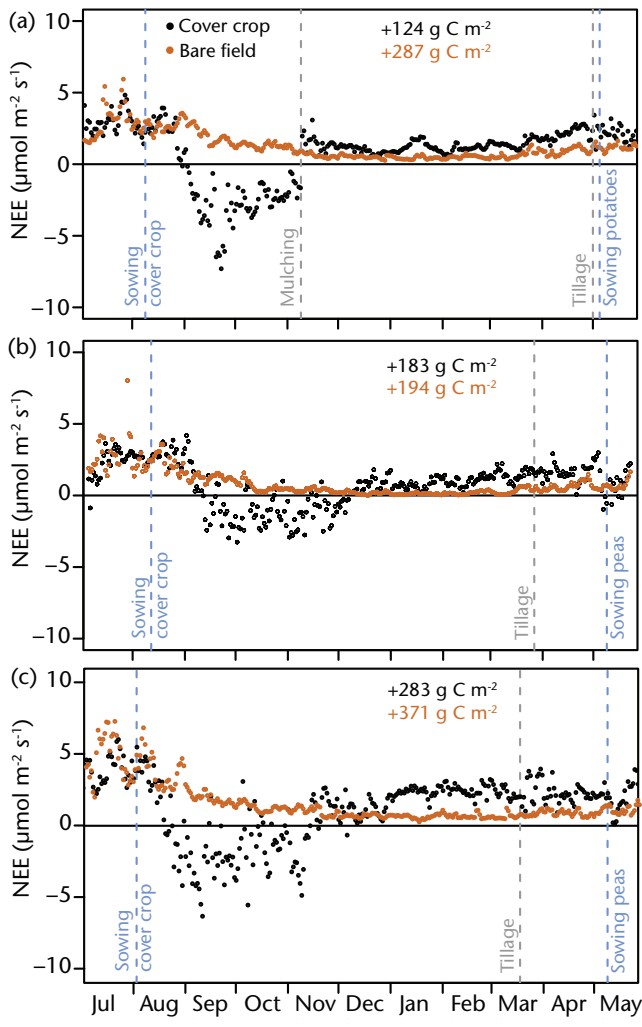

**Figure 5.** Daily average NEE of the three cover crop seasons: (a) 2005/2006, (b) 2009/2010 and (c) 2015/2016 displaying measured data with cover crop and modelled data with a bare field. Measured NEE was measured with an EC system while modelled NEE was simulated with the model SPA-Crop. Vertical lines indicate sowing, tillage and mulching dates. Numbers in the top right corner of each subfigure are cumulative NEE of the field with cover crop in black and of the bare field in brown.





t ha$^{-1}$ would compensate the C losses without any further slurry applications if we assume no increase in $R_{eco}$. The regular application of solid manure could also reduce the amount of mineral fertilisers applied to the field because substantial amounts of N, phosphorus pentoxide ($P_2O_5$), potassium oxide ($K_2O$) and magnesium (Mg) would be supplied by the solid manure (for N approximately half and in all other cases close to the needs as given by the fertilisation plan (Landwirtschaftliche

Beratungszentrale Lindau LBL, 2005) averaged over all crop seasons). The application of compost instead of solid manure should be considered if not enough solid manure is produced by the farm. We estimate that 28.6 t ha$^{-1}$ of compost would be needed to compensate the average annual C losses assuming that the net fluxes of compost are similar to manure. Also, in the case of compost, large fractions of the N, $P_2O_5$ and $K_2O$ needs would be met. On the other hand, Mg would be overfertilised. This is however only a rough estimate because the composition of compost and manure can vary substantially. Furthermore,

the manure amount needed to compensate C losses should be rather seen as a lower limit because several studies in Switzerland have shown that the C loss reduction can be much less then the C input through manure (10 to 30 % of C inputs; Leifeld et al., 2009; Oberholzer et al., 2014; Maltas et al., 2018), which likely also applies to compost. Including ley in the crop rotation could also be considered to compensate C losses. According to Maltas et al. (2018) green manure or cereal straw application can also be effective measures to prevent or reduce soil degradation, while solid manure has, however, the highest C loss reduction

efficiency (compost was not included in the study).

**Table 3.** Nutrient requirements and input: Average annual need based on the fertilisation plan (Landwirtschaftliche Beratungszentrale Lindau LBL, 2005), the input of the same nutrients with the application of 15.8 t ha$^{-1}$ of solid manure (based on the average concentrations of solid manure given in Table D1) and the application of 28.6 t ha$^{-1}$ of compost (based on concentrations from Landwirtschaftliche Beratungszentrale Lindau LBL, 2005). For N an efficiency of 60 % was assumed for manure and for compost as required by the regulations of PEP in the case of farmyard manure (Amaudruz et al., 2014).

|  | Average annual need | 15.8 t ha$^{-1}$ solid manure | 28.6 t ha$^{-1}$ compost |
|---|---|---|---|
| $C_{org}$ (g m$^{-2}$) | 174.0 | 174.0 | 174.0 |
| N (g m$^{-2}$) | 12.7 | 5.4 | 6.0 |
| $P_2O_5$ (g m$^{-2}$) | 7.5 | 6.8 | 5.7 |
| $K_2O$ (g m$^{-2}$) | 14.2 | 11.3 | 8.1 |
| Mg (g m$^{-2}$) | 1.7 | 1.7 | 4.4 |

## 4 Conclusions

The combination of direct eddy covariance measurements and management records provided a unique dataset to study the long-term C budget of the crop field over thirteen years. The field was managed under the regulations of the Proof of Ecological Performance (PEP) regulations that shift the focus from a purely economical focus to a more ecological one. Our goal was to assess whether the PEP regulations resulted in an improved C budget.

Our study showed that the crop field was a source of C of 1674 g C m$^{-2}$ over thirteen years (129 g C m$^{-2}$ per year), which was also confirmed by changes in the soil C stock in the top 30 cm. The loss corresponds to a soil C stock loss of 15.7 to 18.5 % over these thirteen years of study.

Overall, NBP of most crop seasons was positive (i.e., the field lost C), while the C loss during pea seasons was the largest. Liquid manure had a too small C content to compensate C losses of a whole crop season. Contrastingly, solid manure imported similar C amounts into the ecosystem as the C uptake through NEE of the cereal and rapeseed crops.

The field was a net C source during cover crop seasons, but model simulations showed that the source was smaller than if the field would have been left bare between the autumn and spring before a summer crop was sown.

Managing the field under the regulations of PEP did not result in a long-term C sink. However, some aspects of the regulation seem to improve the C budget of croplands. Even though the application of slurry had very little influence on the C budget, fertilisation with solid manure and the sowing of cover crops during fallow periods provide a potential means to close the C budget of this crop field. More effort than only applying PEP is necessary to reach not only an N-neutral but also a C-neutral budget and to meet Switzerland's NDC. The more frequent application of solid manure or compost should be considered to at least partly compensate the C losses with the side effect of reducing the need for mineral fertilisers.

*Data availability.* Gap filled observational NEE, SPA-Crop modelled NEE, soil C and N concentrations, harvest exports, sowing and fertiliser inputs and ancillary meteorological and soil data will be made available under https://doi.org/10.3929/ethz-b-000260058.





## Appendix A: Carbon budget tables

**Table A1.** Seasonal carbon budget expressed as cumulative net biome production ($NBP_{cum}$) and its contributing terms of the 15 full crop seasons between 2004 and 2016 (units: g C m$^{-2}$). A season is defined as the period from the sowing of the current crop until the sowing of the following crop. $NEE_{cum}$ is the cumulative net ecosystem exchange, $E_{harvest}$ is the C export through harvest, and $I_{fertiliser}$ and $E_{sowing}$ are the C imports through organic fertilisation and sowing, respectively. The sums over all crop seasons are also given.

| Season | Crop | $NEE_{cum}$ | $E_{harvest}$ | $I_{fertiliser}$ | $I_{sowing}$ | $NBP_{cum}$ |
|---|---|---|---|---|---|---|
| 16 Oct 2003−28 Sep 2004 | Wheat | −326 | 449 | 0 | −7 | 116 |
| 29 Sep 2004−08 Aug 2005 | Barley | −226 | 401 | 0 | −5 | 170 |
| 09 Aug 2005−04 May 2006 | Cover | 131 | 0 | −148 | 0 | −17 |
| 05 May 2006−18 Oct 2006 | Potato | −17 | 0 | 0 | −15 | −32 |
| 19 Oct 2006−27 Jul 2007 | Wheat | −150 | 401 | 0 | −8 | 243 |
| 28 Jul 2007−06 Oct 2008 | Rapeseed | −118 | 177 | 0 | 0 | 59 |
| 07 Oct 2008−11 Aug 2009 | Wheat | −286 | 407 | 0 | −7 | 114 |
| 12 Aug 2009−08 May 2010 | Cover | 190 | 0 | −123 | 0 | 67 |
| 09 May 2010−14 Oct 2010 | Peas | 185 | 35 | 0 | −4 | 215 |
| 15 Oct 2010−23 Sep 2011 | Wheat | −395 | 424 | −16 | −7 | 6 |
| 24 Sep 2011−03 Sep 2012 | Barley | −360 | 397 | −25 | −7 | 5 |
| 04 Sep 2012−18 Oct 2013 | Rapeseed | −212 | 204 | −25 | 0 | −33 |
| 19 Oct 2013−28 Sep 2014 | Wheat | −264 | 454 | −25 | −8 | 157 |
| 29 Sep 2014−02 Aug 2015 | Barley | −251 | 376 | 0 | −5 | 120 |
| 03 Aug 2015−08 May 2016 | Cover | 293 | 0 | −229 | 0 | 64 |
| 09 May 2016−11 Oct 2016 | Peas | 407 | 3 | 0 | −3 | 407 |
| Sum | All crops | −1400 | 3728 | −591 | −76 | 1661 |





**Table A2.** Annual carbon budget expressed as cumulative net biome production ($NBP_{cum}$) and its contributing terms for the thirteen crop years between 2003 and 2016 (units: g C m$^{-2}$). A crop year starts here on 16 October of one year and ends on 15 October of the next year. This date was used because the first crop was planted on 16 October 2003. $NEE_{cum}$ is the cumulative net ecosystem exchange, $E_{harvest}$ is the C export through harvest, and $I_{fertiliser}$ and $E_{sowing}$ are the C imports through organic fertilisation and sowing, respectively. The total sum, annual average and standard error of each term is also given.

| Season | Crop | $NEE_{cum}$ | $E_{harvest}$ | $I_{fertiliser}$ | $I_{sowing}$ | $NBP_{cum}$ |
|---|---|---|---|---|---|---|
| 2003/2004 | Wheat | −351 | 449 | 0 | −7 | 91 |
| 2004/2005 | Barley | −359 | 401 | 0 | −5 | 37 |
| 2005/2006 | Cover/potato | 286 | 0 | −148 | −16 | 122 |
| 2006/2007 | Wheat | −150 | 401 | 0 | −8 | 243 |
| 2007/2008 | Rapeseed | −125 | 177 | 0 | 0 | 52 |
| 2008/2009 | Wheat | −371 | 407 | −8 | −7 | 21 |
| 2009/2010 | Cover/peas | 438 | 35 | −115 | −4 | 355 |
| 2010/2011 | Wheat | −433 | 424 | −16 | −7 | −32 |
| 2011/2012 | Barley | −331 | 397 | −25 | −7 | 35 |
| 2012/2013 | Rapeseed | −185 | 204 | −25 | 0 | −6 |
| 2013/2014 | Wheat | −274 | 454 | −25 | −8 | 148 |
| 2014/2015 | Barley | −365 | 376 | 0 | −5 | 6 |
| 2015/2016 | Cover/peas | 833 | 3 | −229 | −4 | 603 |
| Sum | | −1387 | 3728 | −589 | −78 | 1674 |
| Average | | −107 | 287 | −45 | −6 | 129 |
| Standard error | | 107 | 49 | 20 | 1 | 50 |





## Appendix B: Soil carbon and nitrogen

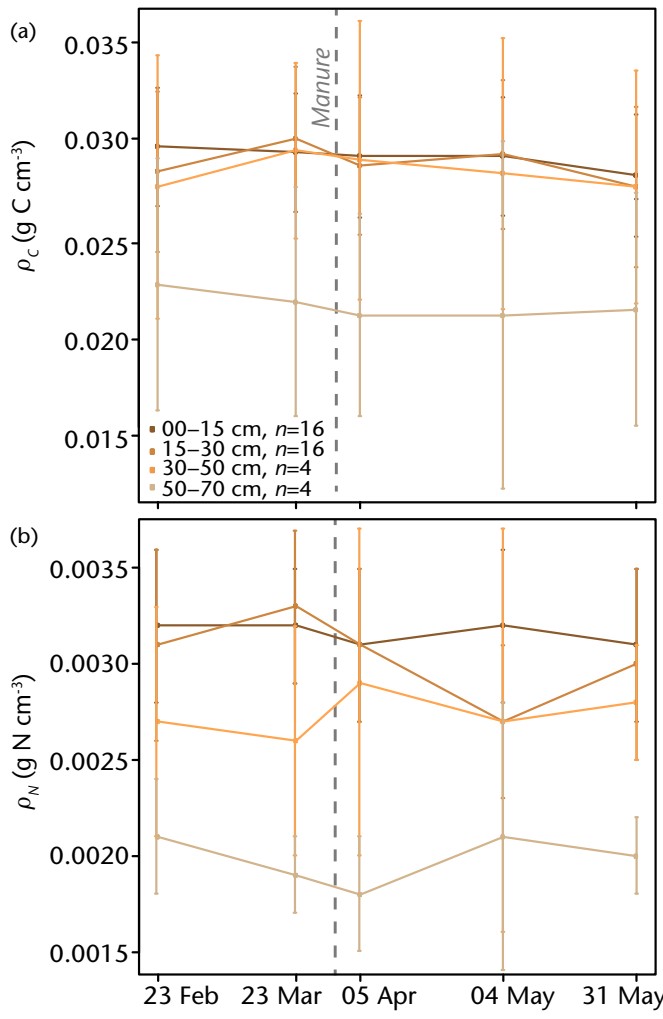

**Figure B1.** Average soil carbon (a) and nitrogen (b) densities ($\rho_C$ and $\rho_N$, respectively) in five different soil layers and on two days before and three days after the application of liquid manure in 2017. Standard deviations are shown as error bars. The grey dashed line indicates the day of manure application. The number of samples ($n$) included in the averages is given in the legend.





## Appendix C: $NEE_{cum}$ and $NBP_{cum}$ with seasons defined by ploughing

**Figure C1.** Crop season specific cumulative net ecosystem exchange ($NEE_{cum}$) and net biome productions ($NBP_{cum}$) with a season defined as the time range between first ploughing after the harvest of the previous crop to the first ploughing after the harvest of the current crop. Each symbol stands for one crop season.



## Appendix D:  Fertilizer inputs

**Table D1.** Average nutrient concentrations (per dry matter) of liquid and solid manure. The liquid manure data of 2017 are based on samples from 31 March 2017 and include all variables while for the average over all liquid manure samples between 2002 and 2017 only dry mass, C and N data are available. The solid manure data are based on 5 samples on 24 January 2006. The number of samples included in the average is given as $n$.

|  | Liquid 2017 | Liquid 2002−2017 | Solid 2006 |
| --- | --- | --- | --- |
| $n$ | 2 | 22 | 5 |
| Dry mass (%) | 2.1 | 2.4 | 25.0 |
| C/N ratio | 5.4 | 4.0 | 18.9 |
| $C_{org}$ (g kg$^{-1}$) | 412.5 | 324.0 | 440.0 |
| N (g kg$^{-1}$) | 76.6 | 81.0 | 22.9 |
| $P_2O_5$ (g kg$^{-1}$) | 19.9 | n.a. | 17.2 |
| $K_2O$ (g kg$^{-1}$) | 109.7 | n.a. | 28.6 |
| Mg (g kg$^{-1}$) | 5.5 | n.a. | 15.4 |
| Ca (g kg$^{-1}$) | 15.6 | n.a. | 4.2 |

*Author contributions.*  CE designed the study, conducted most parts of the analysis, wrote and revised the manuscript. AW and CE designed and conducted the slurry application study in 2017. CA contributed parts of the field management and manure data. LH was involved in processing the eddy covariance measurements. WE conducted the uncertainty analysis of the elemental analysers and supported CE during
5   all parts of the study. All co-authors were involved in writing and contributed to the study with feedback and critique.

*Competing interests.*  The authors declare that they have no conflict of interest.

*Acknowledgements.*  This project was funded by the Swiss National Science Foundation (SNF) grant 146373. We thank the farmers Daniel and Walter Ingold for the management of the field and providing information on the management, Agroscope for providing access to the field site and infrastructure, involved technicians (Peter Plüss, Thomas Baur, Philip Meier, Florian Kaeslin, Patrick Koller, Ivo Beck and Paul





Linwood) and student helpers (Seyhan Kâhya, Anja Taddei, Ewa Merz, Eva Penz) for maintaining the site, helping with the measurements and summarising field information. We also thank Karin Grassow for taking the soil samples in 2004 and Annika Ackermann for performing the C and N elemental analyses as well as Christoph Bachofen for his statistical advice.





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
