# Peer review of "Integrated management of a Swiss cropland is not sufficient to preserve its soil carbon pool in the long-term"

_Biogeosciences, 2018_

## Referee Comment (RC1) · Anonymous Referee #1 · 17 May 2018

The manuscript describes the long-term C budget of a Swiss cropland field over thirteen years. CO2 fluxes were measured by eddy covariance and import or export through harvest, organic amendments and seeds were registered by the farmer (and the C content analysed). There are to my knowledge only very few comparable studies in croplands and therefore this study is timely. In particular because there is political interest in the potential of C sequestration of croplands (launched at the COP in Paris; 4 per mille initiative). The C budget approach in croplands is rather sensitive to errors, and the authors estimate these errors based on literature references. The results of the study are compared to a more traditional approach of changes in soil C stocks before and after the thirteen years. These results compare rather well with the C budget.

The paper is well-written and the experiments and data analysis is sound. My main remarks are on the context and on the implications of the study. The losses (in soil C) are rather large at 1.2 Mg C ha-1 y-1. It should be noted that such high losses are to a large extent a result of the initial conditions. The authors mention an 8 year rotation with 3 years of temporary grassland. This rotation is likely to reach a much higher soil C stock than the cropland rotation that followed. I would appreciate if this could be mentioned in the discussion. After all, a continuous loss of 1.2 mg C ha-1 y-1 seems unlikely, given that most croplands contain round 50 Mg C ha-1 in the top 30 cm. Also for the context, there is a recent literature review on the potential of C sequestration by conservation agriculture (Chenu et al in press). Some of the measures (e.g. cover crops) are also discussed in this review and are reported to sequester C. I would appreciate your views on this paradox. Finally, you mention the application of manure as a measure to compensate C losses in the framework of the GHG reporting (page 17, lines 5-10 and Conclusion lines 18-19. There is some discussion on the role of organic amendments for the sequestration of atmospheric CO2. Powlson et al (2011) argue that amendments transfer C from one location to another, but do not sequester CO2 from the atmosphere. I believe Chenu et al (in press) also address this issue.

Chenu, C., Angers, D.A., Barré, P., Derrien, D., Arrouays, D., Balesdent, J. (in press) Increasing organic stocks in agricultural soils: Knowledge gaps and potential innovations. Soil and Tillage Research Powlson, D.S., Whitmore, A.P., Goulding, K.W.T. 2011.Soil carbon sequestration to mitigate climate change: A critical re-examination to identify the true and the false. European Journal of Soil Science, 62 (1), pp. 42-55.

---

## Referee Comment (RC2) · Anonymous Referee #2 · 21 Jun 2018

The paper is a very important contribution to the discussion on climate mitigation in agriculture even though it covers data only from one site. The site Oensingen has been monitored since 2003 including eddy covariance and soil carbon stocks inventories. Most important for the net C balance of the site is the import of C via organic amendments and the export with harvest. These fluxes are often ignored or insufficiently quantified. In contrast this study presents the full balance including all fluxes. It is written very well and provides a unique data set and valuable conclusions on cover crops and organic fertilisation effect on the C balance of croplands.

There are only two major aspects that should be considered to improve the paper:

[Figure]

1.) Cover crops are presented (e.g. in table 2 and Fig. 4) similar to other crop types even though they are not similar, since they are only grown during autumn and winter season. This should be made clear to the reader at any point (e.g. by adding "winter season" in figure legends). Also on page 14, in l. 12 I recommend to rephrase to: "Their relatively large C loss to the atmosphere was thus a result of the winter growing season and not of the crop type". On p. 14 l. 3: Do not merge peas and cover crops in one sentence since they are not comparable. P. 15 l. 14: Rephrase to "During the winter season with cover crops there was always a net C loss. This loss….". Add on p. 10 in l. 15 "of net CO2 loss during winter season". I would also recommend changing the order of crop types in all relevant figures and tables and putting cover crops separately at the end. 2.) There is a discussion on organic fertilisers to compensate soil C losses in agriculture since they can be so called external C inputs (see Powlson, Whitmore et al. European Journal of Soil Science 2011). If the organic fertiliser is made of biomass that is mainly produced outside the field side where it is applied, we do not gain a climate mitigation effect. Thus, it would be important to estimate from straw and yield export of the Oensingen site how much of this exported C is fed to animals or mixed with manure and afterwards returned to the field site. Is it a closed cycle? You recommend increasing the use of organic fertilisers (e.g. p. 18. L. 18). However, you do not mention how much organic fertiliser is produced from biomass from the site or maybe available at regional scale. Also the recommendation for more compost application as organic amendments (p.17. l.5) should be encompassed with data on how much compost in regionally available.

Minor remarks P. 1, l.2: "intensively managed" P. 1, l.7: Mention here that NEE derived NBP is not equal to the net C balance since non CO2 fluxes (DOC etc.) have to be included but are often very small. P.2, l.27-30 can be removed since they only contain large lists of existing studies (blowing the reference list) without referring to the content and finding of these studies. P.5, l. 1: The type and company that produced the camera is no required information to understand or potentially reproduce this study. Please remove it. P.5, l.25: Please provide the physically plausible range that was
used as quality criteria (-50 to +50?). P.5, l. 31: How was $CO_2$ storage measured or calculated? Please explain. P.7, l. 1: How was the u* filter determined? Which criteria were used to keep it dynamic? P.7, l. 28: Provide the depth for which bulk density was determined. P. 7, l. 30: Explain why soil sampling was interfered by liquid manure application and what could be the consequences for the obtained soil C and N data. P. 8 Eq. 1: Add "+N" with N comprising all non $CO_2$ losses as mentioned on the next page. Is there any estimation of the DOC export flux for the site in order to show that it is small and can be ignored? P. 9, l. 8: Soil-Plant-Atmosphere Crop Model P. 10, l. 3: Did the season starts with sawing or with January? How is it defined? P. 10, l. 8: Voluntary growth??? Fig. 3: Please add the estimated soil carbon density for the topsoil for the 2004 sampling in this figure. P. 12, l. 2-3: Please provide uncertainties for the relative C loss numbers. P. 12, l. 6: The number in brackets "(138. . ." is unclear what is it referring to. Please also consider to compare your mean NBP with data from Kutsch et al 2010 and Schulze et al 2009 or data from cropland soil inventories provided by Ciais et al. 2010, GCB. P. 13, l. 9: More work on the uncertainties of the flux components would very much improve the paper. Could you provide estimates for the uncertainties instead of only stating "they can be assumed to be similar"? P. 15, l. 4: "was again comparable" – comparable to what? P. 15 l. 32: "promised" instead of "assumes" ?! Tab. 3: This is a table which I do not understand. First, Corg is no nutrient and should be removed from this table. Where do the amounts of solid manure and manure come from? Is it annual? If yes, the unit should be Mg ha-1 a-1. P. 18: l. 7: Please take into account the uncertainty of the numbers and do not provide them with one digit (since they cannot be determined with such a high accuracy).

---

## Author Comment (AC1) · 11 Jul 2018

**Response to interactive comments of reviewer 1 (bg-2018-205-RC1)**

We provide the reviewer's comments and critique in blue and provide our response in black. All pages and lines are from the revised manuscript unless otherwise stated.

The manuscript describes the long-term C budget of a Swiss cropland field over thirteen years. CO2 fluxes were measured by eddy covariance and import or export through harvest, organic amendments and seeds were registered by the farmer (and the C content analysed). There are to my knowledge only very few comparable studies in croplands and therefore this study is timely. In particular because there is political interest in the potential of C sequestration of croplands (launched at the COP in Paris; 4 per mille initiative). The C budget approach in croplands is rather sensitive to errors, and the authors estimate these errors based on literature references. The results of the study are compared to a more traditional approach of changes in soil C stocks be- fore and after the thirteen years. These results compare rather well with the C budget. The paper is well-written and the experiments and data analysis is sound.

Thank you very much for this positive assessment.

My main remarks are on the context and on the implications of the study. The losses (in soil C) are rather large at 1.2 Mg C ha-1 y-1. It should be noted that such high losses are to a large extent a result of the initial conditions. The authors mention an 8 year rotation with 3 years of temporary grassland. This rotation is likely to reach a much higher soil C stock than the cropland rotation that followed. I would appreciate if this could be mentioned in the discussion. After all, a continuous loss of 1.2 mg C ha-1 y-1 seems unlikely, given that most croplands contain round 50 Mg C ha-1 in the top 30 cm.

We added a sentence to the discussion that the preconditions of the field likely enhanced the C losses:
"The loss strength, however, was likely influenced by the arable-ley rotation, which was used at the field until the late 1990s and which is expected to reach a higher soil C stock than the crop rotation that was used afterwards." (P.11, l. 10-11)

Also for the context, there is a recent literature review on the potential of C sequestration by conservation agriculture (Chenu et al in press). Some of the measures (e.g. cover crops) are also discussed in this review and are reported to sequester C. I would appreciate your views on this paradox.

Chenu, C., Angers, D.A., Barré, P., Derrien, D., Arrouays, D., Balesdent, J. (in press) Increasing organic stocks in agricultural soils: Knowledge gaps and potential innovations. Soil and Tillage Research

Powlson, D.S., Whitmore, A.P., Goulding, K.W.T. 2011.Soil carbon sequestration to mitigate climate change: A critical re-examination to identify the true and the false. European Journal of Soil Science, 62 (1), pp. 42-55.

We added the following sentences to the text:
"In contrast to tropical regions (Powlson et al., 2016), where climate during cover crop seasons is not a limiting factor, the field experienced a net loss of C during the cover crop seasons due to the less favorable climate (colder and less light) on the Swiss Plateau in autumn" (P. 15, l. 11-13)

"In a recent review by (Chenu et al., 2018) the use of cover crops was discussed. Similar to our findings they conclude based on a number of different studies that the use of cover crops is beneficial for soils because it results in higher soil organic C stocks compared to their absence." (P.15, 18-20)

Finally, you mention the application of manure as a measure to compensate C losses in the framework of the GHG reporting (page 17, lines 5-10 and Conclusion lines 18-19. There is some discussion on the role of organic amendments for the sequestration of atmospheric CO2. Powlson et al (2011) argue that amendments transfer C from one location to another, but do not sequester CO2 from the atmosphere. I believe Chenu et al (in press) also address this issue.

In general, we would like to focus on the relevance of C for soil fertility. We rearranged the text to focus more on this aspect. However the potential to compensate C losses to the atmosphere is of course also interesting. We agree that importing manure does not necessarily result in an overall $CO_2$ sequestration because it might be missing somewhere else. This is a very interesting point, however, it would require a complete life cycle assessment which goes beyond the scope of this study.

We made the following changes:
We changed the first sentence in the section to:
"The more frequent use of solid manure could compensate at least partly the C losses of the crop field and decrease or prevent the loss of soil fertility." (P. 15, l. 23-24)

We deleted the sentence:
"However, Switzerlands nationally determined contribution (NDC) to the reduction in greenhouse gas emissions assumes zero emissions from non-forest lands like croplands (NDC, 2017) " (in first submission version P. 15, 31-32)

We added the following paragraph:
"Switzerland's nationally determined contribution (NDC) to the reduction in greenhouse gas emissions lists zero emissions from non-forest lands like croplands (NDC, 2017). Therefore, the C losses should be reduced from a climate change point of view. The use of organic fertilisers could help get closer to the set goal. In the case of CH-Oe2, the grains, peas and potatoes were not used to feed animals on the same farm. However, straw produced on the field at a rate of 78 g C m$^{-2}$ year$^{-1}$ (1013 gC m$^{-2}$ in total during the 13 years of measurements) is used on the farm. If this straw would have been added back to the field (either directly or included in solid manure), it could have compensated a fraction of the C losses over the 13 years. Ammann et al. (2007) studied the C exchange of the neighboring grassland managed by the same farm. Intensive management of the grassland fertilised with a mixture of solid and liquid manure from the same farm resulted in a significant

uptake of C. Because the grassland was a C sink it could have been considered to apply the manure to CH-Oe2 instead to counteract the higher C loss of the arable field. Therefore, we assume that there is a potential to decrease the field´s C losses substantially by increasing the application of the farm's own solid manure. In order to determine if the application of manure would improve the greenhouse gas budget of the cropland as listed by Switzerland's NDC, it would require a complete life cycle assessment which goes beyond the scope of this study" (P. 17, l. 9-21)

---

## Author Comment (AC2) · 11 Jul 2018

**Response to interactive comments of reviewer 2 (bg-2018-205-RC2)**

We provide the reviewer's comments and critique in blue and provide our response in black. All pages and lines are from the revised manuscript unless otherwise stated.

The paper is a very important contribution to the discussion on climate mitigation in agriculture even though it covers data only from one site. The site Oensingen has been monitored since 2003 including eddy covariance and soil carbon stocks inventories. Most important for the net C balance of the site is the import of C via organic amendments and the export with harvest. These fluxes are often ignored or insufficiently quantified. In contrast this study presents the full balance including all fluxes. It is written very well and provides a unique data set and valuable conclusions on cover crops and organic fertilisation effect on the C balance of croplands.

Thank you very much for this positive assessment.

There are only two major aspects that should be considered to improve the paper:
1.) Cover crops are presented (e.g. in table 2 and Fig. 4) similar to other crop types even though they are not similar, since they are only grown during autumn and winter season. This should be made clear to the reader at any point (e.g. by adding "winter season" in figure legends).

We added: "Please note that cover crops were only grown during autumn and winter" to the corresponding figure and table legends. (Figure 4, Table 2 and Figure D1)

Also on page 14, in l. 12 I recommend to rephrase to: "Their relatively large C loss to the atmosphere was thus a result of the winter growing season and not of the crop type". On p. 14 l. 3: Do not merge peas and cover crops in one sentence since they are not comparable. P. 15 l. 14: Rephrase to "During the winter season with cover crops there was always a net C loss. This loss. . ..". Add on p. 10 in l. 15 "of net CO2 loss during winter season". I would also recommend changing the order of crop types in all relevant figures and tables and putting cover crops separately at the end.

We made all the textual changes as suggested by the reviewer:
"Their relatively large C loss to the atmosphere was thus a result of the winter growing season, not of the crop type and was strongly compensated by the application of solid manure. " (P. 14, l. 12-14)
"Peas assimilated less C from the atmosphere than the ecosystem released at the same time, and very little was exported from the field during harvest. Also during winter seasons with cover crops, more $CO_2$ was lost to the atmosphere than was taken up by the ecosystem. " (P. 14, l. 2-5)
"During the winter seasons with cover crops, there was always a net C loss. " (P. 15, l. 4)
"Cover crops were only growing in the autumn resulting in a relatively weak $CO_2$ uptake followed by a relatively long period of net $CO_2$ loss during winter season." (P. 10, l. 12-13)

We also changed the order of the crop species in Figure 4 and C1 and Table 2 so that cover crops appear at the end.

2.) There is a discussion on organic fertilisers to compensate soil C losses in agriculture since they can be so called external C inputs (see Powlson, Whitmore et al. European Journal of Soil Science 2011). If the organic fertiliser is made of biomass that is mainly produced outside the field side where it is applied, we do not gain a climate mitigation effect. Thus, it would be important to estimate from straw and yield export of the Oensingen site how much of this exported C is fed to animals or mixed with manure and afterwards returned to the field site. Is it a closed cycle? You recommend increasing the use of organic fertilisers (e.g. p. 18. L. 18). However, you do not mention how much organic fertiliser is produced from biomass from the site or maybe available at regional scale. Also the recommendation for more compost application as organic amendments (p.17. l.5) should be encompassed with data on how much compost in regionally available.

As also mentioned in the responses to the other reviewer's comments, in general, we would like to focus on the relevance of C for the soil fertility. We rearranged the text to focus more on this aspect. However the potential to compensate C losses to the atmosphere is of course also interesting. We agree that importing manure does not necessarily result in an overall $CO_2$ sequestration because it might be missing somewhere else. This is a very interesting point, however, it would require a complete life cycle assessment which goes beyond the scope of this study. We do not have all the numbers to assess the life cycle of all C adequately. This is a highly interesting research topic on its own and would require an additional study, beyond the scope of this paper. However we provided some information to estimate the potential.

Compost is not a limiting factor in the region. For example, the composting plant Ricoter, which is located relatively close to the field site (in Aarberg, at 45 km distance) produces 200 000 $m^3$ per year recycled soil substrate from compost (approximately 160 000 t). Recycled soil substrate consists to a major part of compost mixed with low-organic substance soils from construction sites. Several other composting plants exist close to the field site. In total, there exist 368 composting plants in Switzerland. At some of the sites, it is even the policy to offer the recycled soil substrate freely to farmers who want to ameliorate their fields (e.g. Allmig near Baar, Switzerland). It is thus more a question on the carbon footprint of transporting such recycled soil substrate to a given site, i.e. a full life cycle impact assessment, not primarily a question of the availability of the material.

We implemented the following textual changes:

We changed the first sentence in the section to:
"The more frequent use of solid manure could compensate at least partly the C losses of the crop field and decrease or prevent the loss of soil fertility." (P. 15, l. 23-24)

We deleted the sentence:

"However, Switzerlands nationally determined contribution (NDC) to the reduction in greenhouse gas emissions assumes zero emissions from non-forest lands like croplands (NDC, 2017) " (in first submission version, P. 15, 31-32)

We added the following paragraph:
"Switzerland's nationally determined contribution (NDC) to the reduction in greenhouse gas emissions lists zero emissions from non-forest lands like croplands (NDC, 2017). Therefore, the C losses should be reduced from a climate change point of view. The use of organic fertilisers could help get closer to the set goal. In the case of CH-Oe2, the grains, peas and potatoes were not used to feed animals on the same farm. However, straw produced on the field at a rate of 78 g C $m^{-2}$ $year^{-1}$ (1013 gC $m^{-2}$ in total during the 13 years of measurements) is used on the farm. If this straw would have been added back to the field (either directly or included in solid manure), it could have compensated a fraction of the C losses over the 13 years. Ammann et al. (2007) studied the C exchange of the neighboring grassland managed by the same farm. Intensive management of the grassland fertilised with a mixture of solid and liquid manure from the same farm resulted in a significant uptake of C. Because the grassland was a C sink it could have been considered to apply the manure to CH-Oe2 instead to counteract the higher C loss of the arable field. Therefore, we assume that there is a potential to decrease the field´s C losses substantially by increasing the application of the farm's own solid manure. In order to determine if the application of manure would improve the greenhouse gas budget of the cropland as listed by Switzerland's NDC, it would require a complete life cycle assessment which goes beyond the scope of this study (P. 17, l. 9-21)

Minor remarks
P. 1, l.2: "intensively managed"

We changed this as suggested. (P. 1, l. 2-3)

P. 1, l.7: Mention here that NEE derived NBP is not equal to the net C balance since non CO2 fluxes (DOC etc.) have to be included but are often very small.

We added the information by changing the sentence on P.1, l. 11-12 to: "Net biome production, representing the overall C budget (assuming carbon leaching to groundwater to be negligible),…"

P.2, l.27-30 can be removed since they only contain large lists of existing studies (blowing the reference list) without referring to the content and finding of these studies.

To some of the studies we refer to also later, and hence in the introduction we feel that being inclusive in the list of existing studies that we considered in our own work is not a bad thing. The Biogeosciences guidelines do not specify an upper limit for references, hence we decided to keep the list of references in the introduction.

P.5, l. 1: The type and company that produced the camera is no required information to understand or potentially reproduce this study. Please remove it.

We removed the type and company name of the camera.

P.5, l.25: Please provide the physically plausible range that was used as quality criteria (-50 to +50?).

The plausible range is already mentioned right at the end of this sentence in brackets: "Fluxes were rejected from further analyses when they were outside a physically plausible range ($\pm50$ µmol m$^{-2}$ s$^{-1}$)" (P.6, l. 13; in first submission version P. 5, l. 26)

P5, l. 31: How was CO2 storage measured or calculated? Please explain.

We added: "$CO_2$ storage in the air layer below the flux measurement height was calculated according to Aubinet et al. (2001) within EddyPro." (P. 6, l. 17-18)

P.7, l. 1: How was the u* filter determined? Which criteria were used to keep it dynamic?

We added the following sentence:
"The u* threshold was automatically determined for each bare soil period and growing period separately within REddyProc by determining the saturation of NEE with u*." (P. 6, l. 23-25)

P.7, l. 28: Provide the depth for which bulk density was determined.

We added: "Additionally, bulk density of the soil was determined for a 4-cm deep core within the top 12 cm of the soil…" (P. 7, l. 18-19)

P. 7, l. 30: Explain why soil sampling was interfered by liquid manure application and what could be the consequences for the obtained soil C and N data.

Sampling dates before and after the application of liquid manure were chosen to test if it would change the soil C and N significantly. We added this information to the text: "Sampling dates before and after the application of liquid manure were chosen to test if it changed the soil C and N significantly." (P. 7, l. 21-22)

On P. 11, l. 2-3 (P. 11, l. 20-21 of first submission version), we already stated that the application of liquid manure did not result in a significant change in C and N in the soil.

P. 8 Eq. 1: Add "+N" with N comprising all non CO2 losses as mentioned on the next page. Is there any estimation of the DOC export flux for the site in order to show that it is small and can be ignored?

We added the term I$_{other}$ comprising all other pathways to Eq. 1.

We added the following text:
"The term I$_{other}$ can be relevant in rice paddies, where methane fluxes are important (Nishimura et al., 2008) and at sites, where substantial losses via volatile organic

compounds (VOC) or dissolved organic carbon losses (DOC) have to be taken into account. At the CH-Oe2 site, however, neither of these fluxes is of relevant magnitude, and $I_{other}$ can be neglected. While VOC emissions (methanol) had been investigated at the nearby CH-Oe1 grassland site (Brunner et al., 2007) and were found to be very small compared to $CO_2$ fluxes, no estimates were done for DOC at CH-Oe2 so far. A dye tracer experiment by Alaoui and Goetz (2008) at CH-Oe2, however, indicated that the high clay content actually limits the leakage to lower soil layers well beyond the ploughing depth, hence we do not account for potential DOC losses. " (P. 8, l. 24-31)

P. 9, l. 8: Soil-Plant-Atmosphere Crop Model

We changed this as suggested. (P. 9, l. 6)

P. 10, l. 3: Did the season starts with sawing or with January? How is it defined?

We defined the crop season on P. 3, l. 32 (P. 5, l. 6 of first submission version). Additionally to this, we added "…(defined from sowing of one crop to the sowing of the following crop)…" to P. 10, l. 1.

P. 10, l. 8: Voluntary growth???

Voluntary regrowth is the term that is usually used for the practice to let the recently harvested crop regrow on the field before ploughing them under in the fall. The term is also an important aspect in ICOS ancillary data (e.g. http://www.icos-belgium.be/files/ICOS%20ancillary%20data%20workshop%20-%20Croplands.pdf). We thus decided to keep this terminology.

Fig. 3: Please add the estimated soil carbon density for the topsoil for the 2004 sampling in this figure.

We changed this as suggested.

P. 12, l. 2-3: Please provide uncertainties for the relative C loss numbers.

We added the requested uncertainties: "Depending on the measurement method, the field lost 15.7 ± 4.0 % (based on NBPcum and soil C stock of top 30 cm in 2004) to 18.5 ± 5.3 % (based on soil C stocks in 2004 and 2017) of C over the 13 years. " (P. 11, l. 6-8)

P. 12, l. 6: The number in brackets "(138. . ." is unclear what is it referring to. Please also consider to compare your mean NBP with data from Kutsch et al 2010 and Schulze et al 2009 or data from cropland soil inventories provided by Ciais et al. 2010, GCB.

We changed the sentence to clarify that the number in brackets corresponds to the annual net biome production:

"Ceschia et al. (2010) studied the annual NBP (138±239 g C m−2 year−1, they call it net ecosystem C budget) and the annual changes in soil C stocks of the top 30 cm (2.4±4.7 % year−1) …" (P. 12, l.1-2)

We also added the results of the other three publications as suggested by the reviewer:
"Kutsch et al. (2010) determined an average annual NBP of 95 ± 87 g C m−2 for 5 crop rotation sites and 2 monoculture sites." (P. 12, l. 5-6)

"On the other hand, a study using on a process-based model and soil C inventories (Ciais et al., 2010) and a study using a combination of ecosystem measurements combined with atmospheric greenhouse gas measurements and an inversion model (Schulze et al., 2009) found an average annual source of $8.3\pm13$ to $13\pm33$ g C m$^{-2}$ year$^{-1}$ and $10\pm9$ g C m$^{-2}$ year$^{-1}$, respectively for croplands. In our study, the …" (P. 12, l. 9-12)

P. 13, l. 9: More work on the uncertainties of the flux components would very much improve the paper. Could you provide estimates for the uncertainties instead of only stating "they can be assumed to be similar"?

We added an estimate of the uncertainties of NBP based on Eq. 1 in the appendix (P. 19, l. 4-21) and add a summary sentence to the main text:
"An uncertainty estimate of NBP calculated with Eq. 1 can be found in Appendix A. In total, the uncertainty adds up to a maximum uncertainty of approximately ±25 % of NBPcum. Buysse et al. (2017) listed in detail the uncertainties involved in the different NBP terms, which added up to an uncertainty of 220 g C m$^{-2}$ over the 12 years of their study (at NBP = 990 g C m$^{-2}$) corresponding to an uncertainty of 22 %." (P. 12, l. 16-19)

P. 15, l. 4: "was again comparable" – comparable to what?

We reworded this sentence the following way: "For peas, NEE was comparable to NBP because the export with the harvest was much smaller than for all other harvested crops." (P. 14, l. 29-30)

P. 15 l. 32: "promised" instead of "assumes" ?!

We prefer the term "lists". (P. 17, l. 9)

Tab. 3: This is a table which I do not understand. First, Corg is no nutrient and should be removed from this table. Where do the amounts of solid manure and manure come from? Is it annual? If yes, the unit should be Mg ha-1 a-1.

We removed $C_{org}$ from the table. We changed the caption and the units to make clear that these are annual values. It is already stated in the caption that the amounts for manure are based on table D1, which is based on lab analysis of manure at our field site.

P. 18: l. 7: Please take into account the uncertainty of the numbers and do not provide them with one digit (since they cannot be determined with such a high accuracy).

We changed this as suggested. (P. 18, l. 7)

---

## Author Comment (AC3) · 11 Jul 2018

**Response to Editor comments**

We provide the Editor's comments and critique in blue and provide our response in black. All pages and lines are from the revised manuscript unless otherwise stated.

While reading your manuscript, it was not 100% clear to me how you dealt with observed changes in bulk density while calculating soil C stocks.

As described on P. 7, l. 28-32 (in first submission manuscript P. 8, l. 4-5), the C densities were calculated for both years (2004 and 2017) by multiplying the average C concentration of a depth layer by the corresponding measured average bulk density. The C densities for the different layers were then multiplied by the layer thickness and summed up to determine the C stock.

Another aspect that I think deserves more discussion is why this site accumulated the soil C before 2003, that it is apparently losing since then? Are there potential legacy effects?

We added a sentence to the discussion that the preconditions of the field likely enhanced the C losses:
"The loss strength, however, was likely influenced by the arable-ley rotation, which was used at the field until the late 1990s and which is expected to reach a higher soil C stock than the crop rotation that was used afterwards." (P.11, l. 10-11)

---

## Author Response (AR2)

**Response to comments of editor of 10 August 2018**

We provide the editor's comments and critique in blue and provide our response in black. All pages and lines are from the revised manuscript.

I have now read your revised manuscript and it confirmed my earlier impression that you we very carefully in addressing the concerns raised by the reviewers.

Thank you for your positive assessment.

I have a few points left, which I would like you to address:

As far as I have seen you never described the statistical methods that you used to test whether the changes in soil carbon density, bulk density and stocks were significant. This also includes the study done in 2017 when you did multiple samplings. Please add a section where your statistical methods are described.

We added the following text to the manuscript (P. 7, l. 33 to P. 8, l. 2): "The statistical analysis of differences in soil C and N vertically and over time was conducted in *R*. Significance of soil bulk density as well as soil C and N concentrations, densities and stocks between 2004 and 2017 was determined with a one-sided t-test. To test whether vertical differences in C and N densities in 2017 were significant a one-way ANOVA with following post hoc test was conducted. To test whether the application of slurry in 2017 resulted in a change in C densities, a two-way ANOVA including interactions of the factors time of sampling and depth with following post hoc tests was conducted."

We now provide p values for all statements including significance (P. 11, l. 9 to P. 11, l. 21):" Soil C densities ($\rho_C$) in the top 12 cm of the field were 0.0355 ± 0.0042 g cm$^{-3}$ (mean ± standard deviation) in 2004 and 10 decreased significantly (p < 0.0001) on average by 18.0 % to 0.0291 ± 0.0031 g cm$^{-3}$ until spring 2017 (average over the top 15 cm and over all measurement days in 2017). The bulk density of the same layer increased insignificantly (*p* = 0.25) from 1.16 ± 0.08 g cm$^{-3}$ in 2004 to 1.21 ± 0.14 g cm$^{-3}$ in 2017. The soil C stock decreased significantly (*p* < 0.0001) on average by 775 g C m$^{-2}$ in the top 12 cm from 4263 ± 507 g C m$^{-2}$ to 3488 ± 374 g C m$^{-2}$. At the same time, N stock changes were not significant over the 13 years (372 ± 53 in 2004, 382 ± 44 g N m$^{-2}$ in 2017, *p* = 0.19). There were no measurements from deeper soil layers available for 2004. However, measurements in 2017 show that C densities did not vary significantly (adjusted *p* = 0.959) in the top 30 cm (Fig. 3). Also ploughing was done in most years to a depth of 30 cm. If we therefore assume that C stocks changed equally over a depth of 30 cm between 2004 and 2017, the soil C stock decreased in the top 30 cm layer on average by 1980 g C m$^{-2}$. This corresponds to an annual average loss of 152 g C m$^{-2}$.

The application of slurry caused such a small C input that it was not only invisible in NBP$_{cum}$ (Fig. 2) but was also not detectable in the soil. Soil C density measurements before and after the application of the slurry in 2017 did not reveal any significant (adjusted *p* > 0.05) changes (Fig. C1)."

You present the results of the soil carbon samplings with error bars showing standard deviations. I suggest to use either 95% confidence intervals or standard errors in the graphs (Fig. 3 and Fig C1) since this better illustrates whether differences are significant or not.

We changed the figures. They now show standard errors instead of standard deviations.

You use several times the term 'voluntary regrowth'. I would personally use the term 'spontaneous regrowth'. If you think that 'voluntary regrowth' is a better term, you can leave it as it is.

The term is an important aspect in ICOS ancillary data (e.g. http://www.icos-belgium.be/files/ICOS%20ancillary%20data%20workshop%20-%20Croplands.pdf). We thus decided to keep this terminology.

[revised manuscript text omitted]